# *Clostridium perfringens* Foodborne Outbreak during an Athletic Event in Northern Greece, June 2019

**DOI:** 10.3390/ijerph16203967

**Published:** 2019-10-17

**Authors:** Kassiani Mellou, Maria Kyritsi, Anthi Chrysostomou, Theologia Sideroglou, Theano Georgakopoulou, Christos Hadjichristodoulou

**Affiliations:** 1National Public Health Organisation (EODY), 15123 Athens, Greece; a.chrysostomou@keelpno.gr (A.C.); t.sideroglou@keelpno.gr (T.S.); t.georgakopoulou@keelpno.gr (T.G.); 2Regional Public Health Laboratory (PEDY) of Thessaly, 41221 Thessaly, Greece; mkiritsi@med.uth.gr (M.K.); xhatzi@med.uth.gr (C.H.); 3Department of Hygiene and Epidemiology, Medical School, University of Thessaly, 41500 Larissa, Greece

**Keywords:** *Clostridium perfringens*, toxin, outbreak, foodborne, mass gathering, athletic event

## Abstract

Background: Food safety is a major public health consideration during athletic events. On 27 June 2019, the Hellenic National Public Health Organization was notified of a cluster of gastroenteritis cases among athletes of four of the 47 teams participating at the Panhellenic Handball Championship for children. Methods: A retrospective cohort study among the members of the four teams was performed. The local public health authority visited the restaurants where common meals took place, amassed information on the preparation of meals, and collected samples of leftovers. Stool samples were tested for *Salmonella* spp. and *Shigella* spp. Results: Consumption of minced beef had a statistically significant association with disease occurrence [RR:8.29 (95%CI 1,31-52,7)]. Samples of meat were found positive for *Clostridium perfringens*. It was documented that the meat was not stored and re-heated as indicated. Stool samples were negative for *Salmonella* spp. and *Shigella* spp. and were not tested for the *Clostridium perfringens* toxin. Conclusion: Specific standards should be kept to prevent outbreaks during athletic events. This was the first time that a foodborne outbreak due to *Clostridium perfringens* was investigated in the country. Laboratory investigation for toxins should be enhanced, especially in foodborne outbreaks where clinical manifestations of cases are found to be compatible with infection caused by a toxin.

## 1. Introduction

On 27 June 2019, the Hellenic National Public Health Organization (EODY) was notified by a general hospital in Northern Greece of a cluster of eight children with gastroenteritis symptoms. The children were male handball athletes who had participated at the Panhellenic Athletic Championship of Handball for Children hosted in a town of 18,000 residents. The event had started on 25 June 2019 and was slated to end on 30 June 2019. EODY worked with the local public health directorate to define the extent of the outbreak, investigate the factors that led to its occurrence, and implement measures.

In total, 47 teams with 590 children from all over Greece had arrived at the town on 25 June to participate in the championship. The population of visitors was much higher, which included escorts, trainers, and families of athletes. Communication with the event organizers and team leaders revealed that the recorded cases belonged to four teams with 71 people in total. All cases developed symptoms on 26 June. The four teams had two common meals—one during their trip to the town and one during their stay there. All the other teams were asked to immediately report any case with similar symptoms.

## 2. Methods

### 2.1. Epidemiological Investigation

A retrospective cohort study was performed on the members of the four teams. Information about symptoms and foods eaten were collected using a structured, self-administered questionnaire. A case was defined as any member of the team (athlete, escort) with diarrhea (> two loose stools in 24 hours) on 26 June. Risk ratios (RRs) (calculated after dividing the attack rate among the exposed by the attack rate among the non-exposed) and 95% confidence intervals (CIs) were calculated for food and beverages consumed on 25 and 26 June 2019. Associations were considered statistically significant at *p* < 0.05. 

### 2.2. Laboratory Investigation of Clinical Samples

Stool samples from cases that visited the general hospital were cultured for *Salmonella* spp. and *Shigella* spp. This is considered routine testing for gastroenteritis cases at the hospital. Further testing for bacteria, viruses, and toxins was requested.

### 2.3. Environmental Investigation

The local public health authority visited the restaurants where common meals took place, amassed information on the preparation of meals, and collected samples of leftovers and water (from the public supply system and bottled water).

Samples of minced meat consumed by the athletes was tested for *Salmonella* spp. (ISO 6579-1:2017) and *Listeria monocytogenes* (ISO 11290-1:2017). Furthermore, enumeration of the samples’ coagulase-positive staphylococci (*Staphylococcus aureus* and other species) (ISO 6888-01:1999/Amd. 1:2003), *Escherichia coli* (ISO 16649-2:2001), and sulfite-reducing *Clostridium* spp. (ISO 15213:2017) was performed.

## 3. Results

Of the 71 members of the four teams, 64 were male athletes (9–17 years) and seven were coaches/escorts (five females and two males). All completed the questionnaire (response rate: 100%). Fifty-eight responders met the case definition (82%). Reported symptoms included diarrhea (100%), abdominal pain (97%), and fatigue (97%).

Univariable analysis showed a statistically significant association of disease occurrence with the consumption of minced beef (RR:8.29 (95%CI 1,31-52,7)) served at the restaurant the teams visited on 26 June for lunch (Table 1). All ill responders reported eating pasta with minced beef meat. Among the people that preferred pasta with grated cheese (without meat), none reported gastro symptoms.

The exact time that each case developed symptoms was not systematically recorded and thus the median interval from the consumption of the implicated meal to illness onset could not be calculated. However, the implicated meal took place at 14:00 on 26 June and symptoms of all cases occurred between 20:00 and midnight the same day.

### 3.1. Laboratory Investigation of Clinical Samples

Stool samples from two cases were cultured negative for *Salmonella* spp. and *Shigella* spp. No further laboratory testing was performed. 

### 3.2. Environmental Investigation

During the environmental inspection, it was identified that cooked minced meat was kept in the fridge and not adequately reheated before being served to the team members. The restaurant’s food workers reported no recent illness. Stool specimens from the food workers tested negative for *Salmonella* spp. and *Shigella* spp.; however, they were not tested for the *Clostridium* spp. toxin, as this is not routine in Greece.

The sample of minced meat was negative for *Salmonella* spp. and *Listeria monocytogenes* and was found to have concentrations of coagulase-positive staphylococci and *Escherichia coli* below detection limits (100 cfu/gr and 10 cfu/gr, respectively). However, the concentration of sulfite-reducing *Clostridium* spp. was high (4,5 × 10^6^ cfu/gr). Identification of multiple *Clostridium* spp. colonies to species level using MALDI-TOF MS (Microflex LT mass spectrometer, Bruker Daltonic GmbH, Bremen, Germany) identified the isolates as *C. perfringens* with score values of 2.207–2.296.

Samples of water from the public supply system and bottled water were found to be negative for microbiological indicators and fulfilled the requirements of the legislation. 

### 3.3. Control Measures

Recommendations were given to the restaurant, such as the need for adequate refrigeration, use of thermometers, and implementation of best practices for cooling and reheating foods, including using stainless steel rather than plastic containers, avoiding stacking containers, and ventilating hot food. A follow-up inspection will soon take place.

## 4. Discussion

Mass gathering athletic events always present a challenge for public health authorities and food safety is a major priority [1,2]. Catering for an unusually high number of people requires strict food preparation guidelines.

The symptoms reported in this outbreak, as well as the time of their onset after consumption of the implicated meal, were consistent with *C. perfringens* infection [3]. 

*C. perfringens* is a Gram-positive, spore-forming, anaerobic bacterium, known as a prime causative agent of foodborne and non-foodborne gastroenteritis [4]. Its dormant endospores are highly resistant to environmental insults, such as heat, draught, and sanitizing agents, and can grow at temperatures from 20 °C to 53 °C, while spores can survive high temperatures (up to 95 °C for 1 h) [5,6,7]. The contamination of meat may occur through contact of carcasses with feces, as well as via cross-contamination by other foods or contaminated surfaces during slaughtering [8]. Most *C. perfringens* foodborne outbreaks are attributed to meat and poultry products [9]. 

Improper cooling and holding temperatures, incomplete cooking of foods, and inadequate reheating and temperature maintenance of meat are recognized as major factors contributing to the development of *C. perfringens* outbreaks [10,11,12]. In the present outbreak, the high concentration of *C. perfringens* in the sample of minced meat strengthens the hypothesis that such practices might have contributed to the occurrence of the outbreak. Environmental investigation showed that the minced meat was cooked, cooled, and served without proper reheating. 

In Greece, no *C. perfringens* outbreaks have ever been detected, even though it is an important pathogen involved in foodborne outbreaks [13]. Apart from the fact that the mildness and short duration of symptoms prevent most people from seeking medical attention, laboratory capacity for the detection of the *C. perfringens* toxin at Greek hospitals is low. Only five (6%) of the 84 public general hospitals in Greece reported laboratory capacity to detect the *C. perfringens* toxin in 2018 (unpublished EODY data). 

Even though the investigation of outbreaks caused by toxins has not been a priority in the past, data show that the overall burden of foodborne diseases due to toxins in the community is high, and proper investigation and implementation of public health measures should not be neglected [14,15,16]. 

## 5. Conclusions

This was the first time that a foodborne outbreak due to *Clostridium perfringens* was investigated by public health authorities in the country. Epidemiological investigation can offer useful information regarding the possible source of the outbreak and guide laboratory and environmental investigation. Laboratory investigation for toxins should be enhanced, especially in foodborne outbreaks where clinical manifestations of the cases are found to be compatible with infection caused by a toxin. Finally, specific standards should be kept for the prevention of outbreaks during athletic events.

## Figures and Tables

**Table 1 ijerph-16-03967-t001:** Results of the univariable analysis of the consumption of common meals, among the four teams, cohort study, Panhellenic Athletic Championship of Handball for Children, Greece, June 2019.

Meals	Exposed			Not Exposed			
	Total	ILL	AR%	Total	ILL	AR% *	RR (95% CI)^†^
**Dinner 25/6/2019**							
Pork schnitzel with mashed potatoes	71	58	81.7	0	0		
Tomato salad	63	50	79.4	8	8	100	0.79 (0.70–0.90)
Cherries	70	57	81.4	1	1	100	0.81 (0.73–0.91)
Bottled water							
**Lunch 26/6/2019**							
Pasta with minced beef	63	58	92.1	8	0	12.5^‡^	8.29 (1.31–52.7)
Tomato salad	59	48	81.4	12	10	83.33	0.98 (0.74–1.29)

* Attack rate; ^†^ RR: Relative Risk (calculated after dividing the attack rate among the exposed by the attack rate among the non-exposed), CI: Confidence Interval; ‡ For the calculation of RR and 95% CI one, not exposed case was added.

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
