# Peer review of "Clostridium perfringens Foodborne Outbreak during an Athletic Event in Northern Greece, June 2019"

_ijerph, 2019, doi:10.3390/ijerph16203967_

Round 1

Reviewer 1 Report

Thank you for asking me to review this paper.  It is well written and provides the reader with adequate detail to interpret the findings.

I can see that it was not routine to test for Clostridium spp but it may be useful to insert a sentence at line 89 stating that the food workers' stools were not tested as this is not routine in Greece.

Otherwise, well done.  

Kind regards, 

Author Response

First of all we would like to thank you for your review.

Regarding your comment "... it was not routine to test for Clostridium spp but it may be useful to insert a sentence at line 89 stating that the food workers' stools were not tested as this is not routine in Greece" your point was taken and the sentence was added as suggested.

Reviewer 2 Report

I read the manuscript with interest. Overall the presentation and the results are fine. I have the following comments that, I think, the addressing of which may help potential readers of this paper in their understanding of the results presented.

1) Why do stool samples were tested for C. perfringens?

2) The authors should provide formula for RR values presented in Table 1.

3) It is not clear how the authors have defined exposed and non-exposed groups.

4) In the table footnote, the authors say that for the calculation of RR and 95% CI one not exposed case was added. This seems very ad-hoc.

Author Response

Thank you for your review and the opportunity to improve the manuscript and clarify some issues for the journal’s readers.

1) Why do stool samples were tested for C. perfringens?

Thank you for your comment. Stool samples were not tested for C. perfringens toxin as in Greek hospitals stool samples from gastroenteritis cases are not routinely tested for toxins. By the time we requested collection of samples in order to be further tested at the Regional Public Health Laboratory for a wider spectrum of pathogens all cases had already been discharged from the hospital due to the short duration of symptoms.

On the other hand, meat samples were tested also for C. perfringens amongst others since they were directly sent to the Regional Public Health Lab.

2) The authors should provide formula for RR values presented in Table 1.

Thanks for your suggestion. The information regarding the formula was added as a footnote in Table 1 and in the text (in 2.1. Epidemiological investigation).

3) It is not clear how the authors have defined exposed and non-exposed groups.

Exposure at any of the food items / beverages offered at the meals was defined based on the self report of the team members on whether they had consumed each one of them or not.

4) In the table footnote, the authors say that for the calculation of RR and 95% CI one not exposed case was added. This seems very ad-hoc.

Thank you for the opportunity to clarify this point. Indeed this is an ad hoc, however common practice. When in one shell of the 2 by 2 Table we have zero and thus RR cannot be calculated  as a compromise we insert in this shell the value of 1 or 0.5 in order to calculate the value of RR. In other words this is sth done for calculation reasons. Here, since none of the non-exposed to minced meat developed symptoms it was not possible to estimate the size of the association of disease occurrence with the specific exposure. Using this method we managed to have a rough estimate of the RR of the consumption of pasta with minced meat.